# Shaping the Future of Digitally Enabled Health and Care

**DOI:** 10.3390/pharmacy9010017

**Published:** 2021-01-12

**Authors:** Maureen Spargo, Nicola Goodfellow, Claire Scullin, Sonja Grigoleit, Andreas Andreou, Constandinos X. Mavromoustakis, Bárbara Guerra, Marco Manso, Nekane Larburu, Óscar Villacañas, Glenda Fleming, Michael Scott

**Affiliations:** 1Medicines Optimisation Innovation Centre, Northern Health and Social Care Trust, Antrim BT41 2RL, UK; maureen.spargo@northerntrust.hscni.net (M.S.); nicola.goodfellow@northerntrust.hscni.net (N.G.); Claire.scullin@northerntrust.hscni.net (C.S.); glenda.fleming@northerntrust.hscni.net (G.F.); 2Fraunhofer Institute for Technological Trend Analysis INT, 53879 Euskirchen, Germany; sonja.grigoleit@int.fraunhofer.de; 3Department of Computer Science Mobile Systems Laboratory (MoSys Lab), University of Nicosia Research Foundation, Nicosia 1700, Cyprus; andreou.andreas@unic.ac.cy (A.A.); mavromoustakis.c@unic.ac.cy (C.X.M.); 4EDGENEERING, Lda, 1600-001 Lisbon, Portugal; barbara@edgeneering.eu (B.G.); marco@edgeneering.eu (M.M.); 5Vicomtech Research Centre, 20009 Donostia-San-Sebastian, Spain; nlarburu@vicomtech.org; 6Clínica Humana, 07010 Palma de Mallorca, Spain; ovillacanas@clinicahumana.es

**Keywords:** digital solutions, medicines management, remote monitoring, patient safety

## Abstract

People generally need more support as they grow older to maintain healthy and active lifestyles. Older people living with chronic conditions are particularly dependent on healthcare services. Yet, in an increasingly digital society, there is a danger that efforts to drive innovations in eHealth will neglect the needs of those who depend on healthcare the most—our ageing population. The SHAPES (Smart and Healthy Ageing through People Engaging in Supportive Systems) Innovation Action aims to create an open European digital platform that facilitates the provision of meaningful, holistic support to older people living independently. A pan-European pilot campaign will evaluate a catalogue of digital solutions hosted on the platform that have been specifically adapted for older people. ‘Medicines control and optimisation’ is one of seven themes being explored in the campaign and will investigate the impact of digital solutions that aim to optimise medicines use by way of fostering effective self-management, while facilitating timely intervention by clinicians based on remote monitoring and individualised risk assessments powered by artificial intelligence. If successful, the SHAPES Innovation Action will lead to a greater sense of self-sufficiency and empowerment in people living with chronic conditions as they grow older.

## 1. Introduction

“It’s not how old you are but how you are old.” Jules Renard. 

In 50 years, there will be twice as many Europeans aged over 65 than there are today and almost three times as many people aged over 80 years [1], a triumph indeed for medicine. Yet, as life expectancy increases, so too does the likelihood of age-related problems, such as reduced mobility, social isolation and multimorbidity. If 20% of a person’s life is classified as ‘unhealthy’ [1], huge improvements are needed to address these issues and support people to live healthier and more active lives as they age. Targeted initiatives and collaborations across Europe have been established in recent years to address the challenges faced by older people and, in essence, to help improve the quality of the ‘how’ in Monsieur Renard’s ‘how you are old’ remark.

Coinciding with the need to support an ageing population is the rising demand on healthcare services across Europe. Shifting the delivery of care from hospitals to the home and community may help alleviate pressures faced by the health service. Effective policies, technologies, education and resources are needed in order to establish optimal ‘at-home’ care systems for older individuals with chronic diseases. For this approach to be effective, individuals also need to feel supported and empowered to take responsibility for their own health. Widespread improvements in health literacy and self-management will be crucial to the success of home-based care delivery models.

Multimorbidity (the coexistence of two or more chronic diseases) and polypharmacy (the use of multiple medicines) increase with age, and so too does the likelihood of medication-related problems (for example, age and number of medicines have been shown to be the main factors involved in drug-drug interactions [2]). Therefore, promoting the safe and effective use of medicines is one way in which our ageing population can be supported to self-manage their conditions and remain healthier for longer. The WHO Global Patient Safety Challenges were established to reduce avoidable harm caused to patients through health care [3]. Although not specifically for older people, the third such global challenge entitled ‘Medication without Harm’ focuses on medication safety and specifically addresses polypharmacy, safer transitions of care and high-risk medication situations—all of which pose a particular threat to the health and wellbeing of older people. Nations have been tasked with reducing medication-related harm by 50% over the next five years [3]. 

In Northern Ireland, a five-year action plan to address the WHO Global Patient Safety Challenge has recently been published [4]. Specific objectives include supporting people to become more involved in decisions regarding their medicines and empowering them to report issues with their medication when they are concerned. Supporting adherence is also a key focus of the plan [4]. The plan also acknowledged that eHealth technologies and the digitalisation of the health service will be crucial to meeting the WHO Global Patient Safety Challenge in Northern Ireland [4]. 

The onset of the COVID-19 pandemic has pushed forward the agenda on digital health services, not only in Northern Ireland but globally. Digital solutions rapidly came to the forefront in the provision of care both locally and across the world to support infection control policies and minimise transmission of COVID-19. Technology permitted services to keep running and support patients remotely. A rapid review of how pharmacy services in Northern Ireland adapted in response to the COVID-19 pandemic showed an increase in the delivery of remote or virtual consultations in both primary and secondary care [5]. The use of adherence support apps (i.e., computer programs that run on mobile devices) and text messaging reminders also helped professionals deliver care to patients remotely [5]. The impact of increased uptake of digital health technologies now needs to be evaluated. Even before the COVID-19 pandemic, the number of digital health technologies had been rising at a rapid pace. The National Institute of Health and Care Excellence in the United Kingdom classifies digital health technologies by function, such as system services (e.g., electronic prescribing systems), active monitoring apps that link with sensors, and wearable devices, that aim to permit the remote monitoring of patients by health care professionals, and sophisticated artificial intelligence programmes that use data to guide diagnostic decisions [6]. 

There is now so much choice available that health care organisations need to be able to ensure the digital technologies they deploy are functional, secure and effective at improving clinical care. Further organisational support and guidance is required on how to adapt services to facilitate the development of digital remote assistance. 

Northern Ireland is a four-star reference site for medicines management in the European Innovation Partnership on Active and Healthy Ageing (EIP on AHA), an initiative established in 2011 that aims to promote active and healthy ageing in Europe [1]. Reference sites have been established across Europe to focus on a comprehensive, innovation-based approach to this cause with the overarching aim of increasing the average healthy lifespan of European Union (EU) citizens by two years by the year 2020 [1]. It is still undetermined whether this goal has been achieved, but efforts are ongoing. 

The Medicines Optimisation Innovation Centre (MOIC), which is hosted by the Northern Health and Social Care Trust (a secondary care provider in Northern Ireland), is dedicated to driving innovation in active and healthy ageing and medicines use. It is currently collaborating with researchers across Europe on a project that aims to shape the future of digitally enabled care.

## 2. SHAPES 

### 2.1. The SHAPES Consortium

The SHAPES (Smart and Healthy Ageing through People Engaging in Supportive Systems) Innovation Action is an EU Horizon 2020 co-funded programme (grant agreement 857159), which is coordinated by the National University of Ireland, Maynooth and involves 36 partner organisations across Europe [7]. The programme officially began in November 2019 and will run for 48 months until October 2023. The outputs of an interconnected multitasking workflow within SHAPES are essentially based on the creation of a central technology platform from which many partner organisations can contribute or deploy a range of practical and reliable digital solutions according to a defined need. Ten work packages reflect a rich diversity of disciplines and methodologies—from the ethnographic study of the lifeworld of ageing individuals to the design and development of a pan-European technological platform and the creation of a Marketplace to bring customers and suppliers together. Each work package will both inform and depend upon the other nine. The ‘spine’ of the project is a pan-European pilot campaign that will support the evaluation of the SHAPES platform and a catalogue of associated digital solutions, as well as evaluate the impact of the SHAPES solution on older people. There are seven pilot themes: smart living environment for healthy ageing at home; improving in-home and community-based care; medicines control and optimisation; psycho-social and cognitive stimulation promoting wellbeing; caring for older individuals with neurodegenerative diseases; physical rehabilitation at home; and cross-border health data exchange [7].

Within each pilot theme, selected digital solutions will be adapted, integrated, deployed and evaluated in different real-life circumstances (e.g., a person with multiple health conditions living at home) wherein they have been perceived to be of benefit. These ‘real-life circumstances’ are defined together with their linked digital solutions as ‘use-cases’, and they have been developed according to specific ‘personas’ based upon ethnographic research conducted within the SHAPES project.

### 2.2. Medicines Control and Optimisation

Pertinent to the topic of digital solutions for medicines management is the ‘Medicines control and optimisation’ pilot theme, wherein the SHAPES platform and selected digital solutions will be used to monitor health condition(s), physiological parameters and medicine adherence. Any changes in an individual’s health status can be identified early and managed appropriately. The anticipated benefit of this highly personalised approach to delivering healthcare is to promote the safer and more effective use of medicines in the home, thus improving the health outcomes and quality of life of the target population. Participants in this pilot theme will be individuals who are over 65 years’ old, live at home and have multiple chronic medical health issues—specifically cardiac, respiratory and endocrine conditions. 

As previously described, use-cases will deploy and evaluate different digital solutions according to the type of support required. Four use-cases will be used to evaluate this pilot theme. They are as follows: supporting multi-morbid older individuals (see Section 2.2.1); in-home decompensation prediction for heart failure patients; advanced telemonitoring of patients with heart failure in the home environment; and advanced telemonitoring of patients with chronic obstructive pulmonary disease (COPD) in the home environment. 

Notably, there is an overlap between the digital solutions deployed within each use-case of the pilot theme, and there is replication of the pilot activities in different geographical locations across Europe. This spread will help to evaluate the applicability, usability and generalisability of the solutions in other socio-technical and organisational settings across Europe.

#### 2.2.1. Example Use Case: Supporting Multi-Morbid Older Individuals

The MOIC will be piloting the use case ‘supporting multi-morbid older individuals’. A diagrammatic summary of the SHAPES system to be deployed in this use case can be viewed in Figure 1. At its most basic functionality, this use case is similar to other telemedicine applications. Users are asked to download the SHAPES app to their smartphone or tablet. The participants will provide daily data on their vital signs via Bluetooth enabled devices, specifically weight scales, glucometers, blood pressure monitors and pulse oximeters. The recorded data is electronically transferred to the SHAPES app and artificial intelligence based predictive models may also run to detect problems, such as risk of heart failure decompensation [8]. Participants can view their clinical parameters via the app and have the ability to enter readings manually if required. A healthcare clinician can then monitor these clinical parameters remotely, detecting any changes to the participants’ normal levels and intervening when needed. An added feature of the system is that the app can be personalised to include medication reminders and tracking according to patient need. Questions about the participant’s health are asked daily. For example, ‘In the last three days, did you cough more or have more phlegm?’. The participant can take an active role in communicating changes to his or her treatment with the system, such as new medicines or blood analyses, thereby providing older people with the tools to inform their own care and bridging gaps at transitions of care.

Furthermore, as well as being monitored remotely by a clinician, the clinical and questionnaire data retrieved from the participants are then uploaded via internet connectivity to the SHAPES platform. SHAPES digital solutions can then use artificial intelligence and predictive modelling techniques to monitor data continually and predict deteriorating diseases (i.e., predict heart failure decompensation and hypoglycaemic or hyperglycaemic events). Prior to such events occurring, clinicians are alerted and are then able to intervene promptly to prevent further deterioration.

## 3. The SHAPES Pilot Methodology

### 3.1. The SHAPES Process

The methodology of the SHAPES pilot campaign has been designed and developed by the SHAPES consortium and is the product of significant effective collaboration between the partners. A detailed and extensive pilot plan has been created for all pilot sites to follow. This plan details a phased approach (See Figure 2) to test the SHAPES platform and associated digital solutions and includes information on how the project will be evaluated and what data collection methods will be employed. 

An extensive design and planning phase for the ‘medicines control and optimisation’ pilot theme, during which the overall SHAPES approach and evaluation methodologies will be adapted to the specific pilot sites, concludes in January 2021.

Before conducting the pilot in the target population under ‘real-life’ conditions, the proposed digital solutions will undergo a co-design and user-testing process to optimise their usability and acceptability to the end-users. Between February and April 2021, mock-ups (i.e., simplified visual representations of the actual design of the digital solution) of the patient- and clinician-facing portals will be presented to people from specific user groups (i.e., the target population, clinicians, supportive services) and feedback on the design and functionality will be sought via a semi-structured interview. These sessions will be audio-recorded, transcribed if necessary, and reported. Technical partners who developed the digital solutions will then consider the user feedback when developing the subsequent prototypes for further hands-on experiment and testing. 

For hands-on experiment and testing exercises, participants from the target population will be provided with working prototypes of the digital solutions, along with the connectable medical devices, to try at home using test data. The participants will watch an introductory video explaining the digital solutions and the support they provide. They will then be trained to use all the features available to them that fit their needs and will be asked to have multiple encounters with the products over several days. The aim of this activity for the user is to challenge the functionality and user-friendliness of the products at their own pace. After the settling in period, the user will be monitored using the solutions, both digitally and, if feasible under social distancing regulations, by an independent observer. Users will also be interviewed about their experience using the SHAPES solutions to determine ease of use, what they liked and disliked about the designs, and how useful the solutions have been to them overall. Hands-on testing exercises will take place between May and August 2021. The findings will inform what final adjustments need to be made to the products before deployment. They may also indicate what support materials are required to ensure the users can interact with and use digital solutions as intended. These could include additional explanatory videos, instruction manuals, help buttons or pop-up windows with usage tips. 

Before deployment to real-life settings, the feasibility of the pilot activities will be tested to check if the planned procedures and methodology will work in the target setting and population. The SHAPES digital solutions will be deployed in controlled environments and small-scale live demonstrations will be conducted during which the research team can evaluate if specific steps in the process are running efficiently without the increased workload of a large-scale deployment of the pilot. As the small-scale demonstrations of the digital solutions will take place under controlled settings, potentially highlighting any connectivity issues or shortfalls related to the technology. The recruitment strategy, in particular, will be an essential focus for the small-scale demonstrations and will inform if any changes in approach are required before the deployment in real-life environments.

By the time that the final phase of the large-scale pan-European pilot of the SHAPES platform and digital solutions is ready to begin, the available SHAPES technologies will be in a mature and user-friendly condition and ready to be used in real-life settings. In the example use case, ‘supporting multi-morbid older individuals’, access to download the SHAPES app will be granted, and smart monitoring devices will be given to 30 participants to use at home for the duration of the pilot. Data to measure the pilot outcomes will be collected digitally through tracked usage of the software, via questionnaires distributed using the apps, and in follow-up interviews and questionnaires with participants. Interventions will be logged and assessed accordingly. Small-scale demonstrations in controlled settings will begin in September 2021, closely followed by deployment to real-life environments. Partners at the University of Nicosia Research Foundation in Cyprus will replicate this entire phased approach in tandem with the MOIC for the ‘supporting multi-morbid older individuals’ use case.

### 3.2. Evaluation of the SHAPES Pilot

The evaluation of the SHAPES pilot will address three overarching questions: does the technology work in a real setting; does this approach have any impact; and what can be learned from the experience of conducting the pilot campaign. The performance of the technology (i.e., the platform and the digital solutions) will be evaluated to determine if all technical tools worked as they should and without any errors, if all system requirements were fulfilled, and if the technology can operate within the local infrastructure. Evaluation of the SHAPES technology will also include a detailed and comprehensive user-experience assessment, the methodology of which is being developed by the usability experts of the SHAPES project.

Evaluation of the impact of the SHAPES pilot at each site will include an assessment of changes in health status and well-being of the participants, the satisfaction of the user with the SHAPES solution in its entirety (i.e., not just the technical components), and any effect on caregivers. At an organisational level, changes to the use of healthcare resources, such as hospitalisations, readmissions, medical appointments, and associated costs will be assessed. Specific evaluation frameworks have been selected for use in the SHAPES pilots. The Monitoring and Assessment Framework for the European Innovation Partnership on Active and Healthy Ageing (MAFEIP) and the Model for Assessment of Telemedicine (MAST) will be used where possible to provide an evidence-based structure to the evaluation. The objectives, key performance indicators, outcomes and measurement tools will be set by each pilot site, however, where possible there will be a degree of data harmonisation within and across pilot themes so that data may be pooled and analysed at scale. 

Finally, the third question will examine the organisation and management of the pilot itself. After each individual pilot activity, any problems or challenges that have arisen will be identified and reflected upon so that improvements to future activities can be made. Interviews with key personnel from the respective pilot sites will be conducted after every pilot activity to identify lessons learned and develop recommendations so that there is shared learning across the sites and the opportunity to build upon previous recommendations.

## 4. Discussion

Navigating the local and national restrictions and social distancing advice imposed by the global COVID-19 pandemic is undoubtedly a substantial challenge for the SHAPES project. The majority of research activities, including recruitment processes, data collection, demonstration of solutions and training will occur remotely, and appropriate measures put in place to protect participants. Depending on the local COVID-19 requirements, researchers may not be able to provide physical help to participants with their devices or demonstrate how tasks are performed. As such, the information provided to participants will need to be detailed and easy to follow. Visual aids will be used to support written instructions and facilitate the learning process.

The necessary digital literacy skills will be required for participants to be eligible to take part in pilots, in particular, those which need interaction from users. For example, in the ‘supporting multi-morbid older individual’ use case, participants will need access to a smartphone or tablet and own a stable internet connection. The user-testing and small-scale demonstrations conducted before the pilots will identify any features or functions that do not align with these eligibility criteria and assist with any adjustment. Support features and materials will be produced accordingly and will be informed by the findings of the mock-up and prototype testing user engagement activities. There is also an opportunity through the pilots and participation with SHAPES to enhance the digital literacy of older users via their engagement with the digital solutions and this topic will be explored during the campaign.

This project will validate the ability of the SHAPES platform and digital solutions to optimise the use of medicines by way of fostering effective self-management in people with multiple medical conditions, while facilitating timely intervention by clinicians based on individualised risk assessments powered by artificial intelligence. The findings of this project will contribute to delivering upon the strategic objective in Northern Ireland to drive innovation and digitalisation of health care, whilst also improving medication-related safety and enhancing the quality of life for older people. 

At a higher level, outputs of the pilot campaign will feed into the broader SHAPES work programme, which aims to create an open European platform that provides meaningful, holistic support to older people living independently and facilitates active engagement care networks and local communities. It is recognised that there will be certain groups of people in the target population who do not yet, and perhaps never will, have the digital literacy skills required of them to engage with the full suite of digital solutions offered by the platform. However, inclusivity and accessibility are a priority for the consortium and innovations that seek to provide technical support to all older people whilst also lowering the barrier to participation are being continuously explored. There is also an opportunity with this project to foster a greater sense of self-sufficiency and empowerment in people living with chronic conditions as they grow older.

## 5. Conclusions

One vision of this global approach to innovation in ageing, health and care is for people to embrace digital solutions when they are younger and at an earlier stage in their disease—thus understanding the nuances of their condition at a personal level and practising responsible self-care. Overall, the SHAPES ambition is to build a new paradigm of health and care delivery in homes and community-based settings that supports and extends the healthy and independent living of older people in Europe. With time, we will distance ourselves from the outdated concept of health professionals treating diseases in silos and move towards a reality of people being supported to manage their own health responsibly in a digitally enhanced society.

## Figures and Tables

**Figure 1 pharmacy-09-00017-f001:**
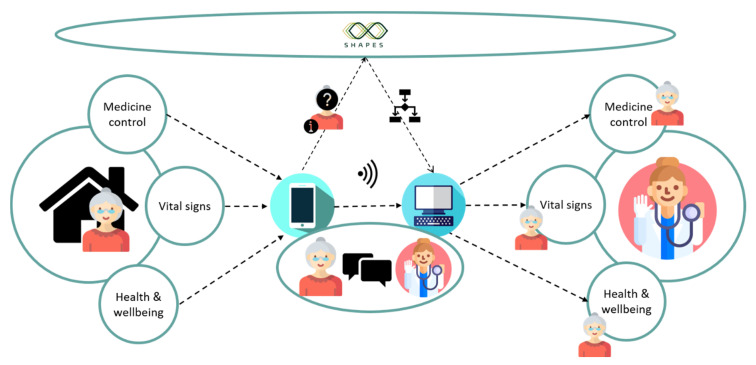
Diagrammatic summary of the SHAPES (Smart and Healthy Ageing through People Engaging in Supportive Systems) system as applied in the ‘supporting multi-morbid older individuals’ use case. The figure illustrates the main features of the patient-held app—medicine control (e.g., adherence tracking and reminders); vital signs (e.g., pulse, blood pressure, oxygen saturation, blood glucose); health and wellbeing (e.g., symptom check). Anonymised data is sent to the SHAPES platform where predictive modelling techniques are deployed to analyse the data and predict deteriorations in disease before they occur. A clinician can monitor a participant’s medicine control, vital signs and health and wellbeing, receive notifications about his or her risk of deterioration via a clinician-facing dashboard and respond appropriately.

**Figure 2 pharmacy-09-00017-f002:**
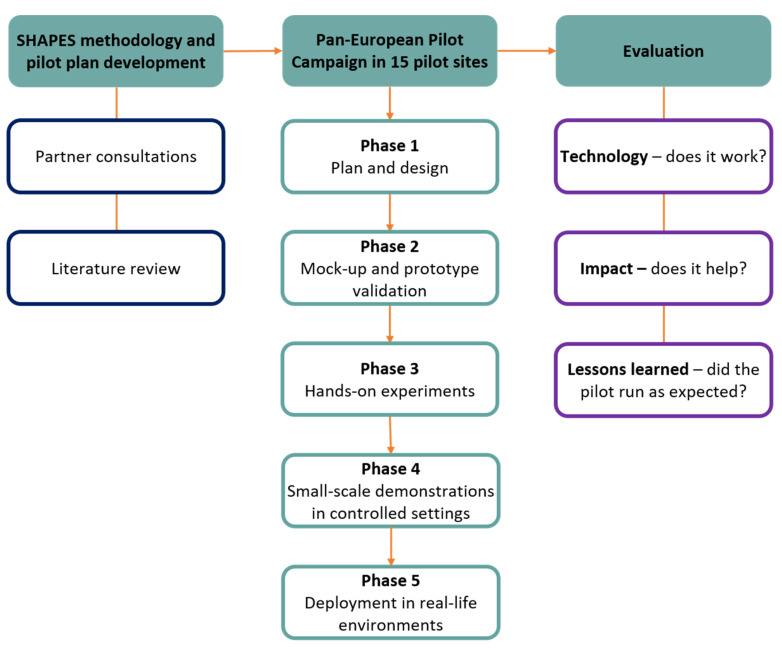
Diagrammatic summary of the SHAPES Methodology. A detailed pilot plan was developed centrally by the consortium and included extensive communication with partners and a review of the available literature. Fifteen pilot sites will undertake activities specific for each use case during the SHAPES pan-European pilot campaign. There are five phases to the pilot: Planning and design; Mock-up and prototype validation; Hands-on experiments; Small-scale demonstrations in controlled settings; and Deployment in real-life environments. The pilot will be evaluated within each site and findings pooled together where possible for an overarching evaluation of the SHAPES Platform and Digital Solutions.

## Data Availability

No new data were created or analyzed in this study. Data sharing is not applicable to this article.

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
