# Peer review of "Shaping the Future of Digitally Enabled Health and Care"

_pharmacy, 2021, doi:10.3390/pharmacy9010017_

Round 1

Reviewer 1 Report

The manuscript deals with very interesting topic, about which authors know a lot. So much that it is not clear in communication, or at least in its first part) whether the future project is announced or the ongoing project is described. What has been already done and what will be done should be more distinct.

Introduction could be more focused. It contains several interesting information but without strong "red line" - at least for the introductional reading without earlier knowledge the SHAPES.

It would be welcome to have a protocol-scheme / activity diagram by phases (methodology design, pilot plan, feasibility of the pilot, primary pilot, small scale demonstration, large scale deployment, large scale pilot, final phase of panEU pilot, replication the pilot,...)

Paragraph 3, last passage: It would be expected some more information about approaches and tools for (big) data analyses.

Since the whole project is oriented to older people, it is suggested to be attentive to jargon – even in the communication with professional readers (e.g. app, sync)

Author Response

Response to Reviewer 1 Comments

Point 1: The manuscript deals with very interesting topic, about which authors know a lot. So much that it is not clear in communication, or at least in its first part) whether the future project is announced or the ongoing project is described. What has been already done and what will be done should be more distinct.

Response 1: We agree that details of the timeline of the project would provide clarity for readers as to what has already been done and what there is still to do. Details about the overall SHAPES project timeline have been inserted on Page 3, Lines 109-110. Details of a preliminary planning phase for the ‘Medicines Control and Optimisation’ pilot theme and when it ended have been added to Section 3 (Page 5, Lines 196-198) to provide readers with more information about what has already been done. Furthermore, details on when the subsequent phases are due to be undertaken have been inserted in the relevant paragraphs (see Page 5, Line 201-202; Lines 220-221; and Lines 246-247.

Point 2: Introduction could be more focused. It contains several interesting information but without strong "red line" - at least for the introductional reading without earlier knowledge the SHAPES.

Response 2: A paragraph has been added early in the introduction (Page 1, Lines 42-49) to provide greater clarity to readers on what the manuscript is about and what SHAPES is attempting to address. The inserted text introduces the concept of shifting healthcare delivery to homes and communities and what changes need to occur for this to be successful. Also mentioned is the need to empower individuals to self-manage their own health. Both these concepts are discussed in the concluding paragraphs of the manuscript and so we believe these changes complement the existing text very well and provide the focus recommended by the reviewer. In addition, the relevance of the rest of the introduction is explained on Page 2, Lines 54 and 55, which hopefully will help focus the existing text even further.  

Point 3: It would be welcome to have a protocol-scheme / activity diagram by phases (methodology design, pilot plan, feasibility of the pilot, primary pilot, small scale demonstration, large scale deployment, large scale pilot, final phase of panEU pilot, replication the pilot,...)

Response 3: Thank you for this helpful suggestion. A second Figure has been added to the manuscript (Page 7, Line 277) to summarise the pilot methodology diagrammatically. Please note also that a more accurate description of the last two phases of the pilot have been provided in the revised document. The ‘small-scale demonstrations’ phase has been further described as taking place in ‘controlled settings’, and rather than describe the final phase as ‘a large scale pilot’ we have more accurately described this phase as ‘deployment in real-life environments.’ (Pages 5-6, Lines 226-248). Clarification has also been provided on the replication of pilot activities in Cyprus Page 6, Lines 248-250.

Point 4: Paragraph 3, last passage: It would be expected some more information about approaches and tools for (big) data analyses.

Response 4: Paragraph 3 has been reformatted to include two sections. The first section details the SHAPES process and 5-phase approach adopted by each pilot site (Pages 5-6, Line 189-245). The second section contains new text and details the approach and tools for evaluation as requested. Please see Page 6, Lines 246 - 275. Finer details about the evaluation have not been provided because the planning of the evaluation methodology is still ongoing.

Point 5: Since the whole project is oriented to older people, it is suggested to be attentive to jargon – even in the communication with professional readers (e.g. app, sync)

Response 5: We agree that jargon should be avoided in all aspects of the project, including our communication with professional readers. Therefore, the term ‘apps’ has been explained as ‘computer programs or software applications designed to run on mobile devices’ on first mention, Page 1, Lines 77-78 and left as is thereafter. The term ‘sync’ on Page 4, Line 160 has been replaced with the term ‘electronically transferred’. Please also note that for clarity, the term ‘their app’ has been changed to ‘the SHAPES app’ on Line 161.

Reviewer 2 Report

I found this article titled, “Shaping the future of digitally-enabled health and care” by Spargo, et. al. a most pleasant and informative read. I offer the following recommended edits and comments for consideration by the authors:

Page 1, Line 34: Recommend the insertion of a colon after the first citation in order to connect the first sentence with the following sentence fragment. The revises sentence should read as follows: “ . . . as many people aged over 80 years [1]: a triumph indeed for medicine.”

Page 1, Line 42: Recommend defining “Multimorbidity and polypharmacy” as these terms may not be easily recognizable by the average Reader, or may have differing definitions among Readers, depending on their backgrounds.

 Page 1, Lines 42-46: Recommend dividing this very long sentence into two sentences.

Page 2, Line 60: Although the authors are consistent in their use of the term “Covid-19” recommend they consider changing this abbreviation to “COVID-19” since the ‘CO’ stands for ‘corona,’ ‘VI’ for ‘virus,’ ‘D’ for disease, and the ‘19’ stands for the year the most recent pandemic began (2019).

 Page 2, Line 60: Recommend the deletion of the word “somewhat” as this adds unnecessary ambiguity to the sentence.

Page 2, Line 80: Recommend the insertion of a hyphen between the words “four” and “star” for clarity. New term should read “four-star”

Page 3, Line 141: The acronym “MIOC” is not fully spelled out prior to its first use. Recommend this acronym be fully spelled out before it is used in the manuscript.

Page 4, Line 160: Recommend replacing the word “is” with “are” as the word ‘data’ is plural (correct noun / verb agreement).

 Page 4, Lines 165-174: This illustration and description of the illustration is very well done and graphically describes to the Reader what the authors are attempting to explain. Well-done.

Page 5, Line 210: Recommend the insertion of a hyphen between the words “small” and “scale” for improved accuracy. The new words should read “small-scale”

Page 5, Line 214: I do not understand the term “teething problems” as it is used in this sentence. Recommend the authors replace this term with a more understandable term or phrase that better conveys what they seek to communicate.

Page 6, Line 250: Recommend replacing the words “medicines use” with “the use of medicines” for improved understanding. The new sentence fragment should read “ . . . digital solutions to optimise the use of medicines by way of fostering . . . “

Author Response

Response to Reviewer 2 Comments

Point 1: Page 1, Line 34: Recommend the insertion of a colon after the first citation in order to connect the first sentence with the following sentence fragment. The revises sentence should read as follows: “ . . . as many people aged over 80 years [1]: a triumph indeed for medicine.”

Response 1: Done. See Page 2, Line 34.

Point 2: Page 1, Line 42: Recommend defining “Multimorbidity and polypharmacy” as these terms may not be easily recognizable by the average Reader, or may have differing definitions among Readers, depending on their backgrounds.

Response 2: Definitions for multimorbidity and polypharmacy have been inserted on Page 2, Lines 50 and 51.

Point 3: Page 1, Lines 42-46: Recommend dividing this very long sentence into two sentences.

Response 3: Done. See Page 2, Lines 50-55.

Point 4: Page 2, Line 60: Although the authors are consistent in their use of the term “Covid-19” recommend they consider changing this abbreviation to “COVID-19” since the ‘CO’ stands for ‘corona,’ ‘VI’ for ‘virus,’ ‘D’ for disease, and the ‘19’ stands for the year the most recent pandemic began (2019).

Response 4: Done. See Page 2; Lines 70-87 (four uses of the term COVID-19) and Page 7, Lines 290 and 293.

Point 5: Page 2, Line 60: Recommend the deletion of the word “somewhat” as this adds unnecessary ambiguity to the sentence.

Response 5: Done. See Page 2, Line 70.

Point 6: Page 2, Line 80: Recommend the insertion of a hyphen between the words “four” and “star” for clarity. New term should read “four-star”

Response 6: Done. See Page 2, Line 92.

Point 7: Page 3, Line 141: The acronym “MIOC” is not fully spelled out prior to its first use. Recommend this acronym be fully spelled out before it is used in the manuscript.

Response 7: The acronym ‘MOIC’ is first used on Page 2, Line 99 where it is spelled out.

Point 8: Page 4, Line 160: Recommend replacing the word “is” with “are” as the word ‘data’ is plural (correct noun / verb agreement).

Response 8: Done. See Page 4, Line 173.

Point 9: Page 4, Lines 165-174: This illustration and description of the illustration is very well done and graphically describes to the Reader what the authors are attempting to explain. Well-done.

Response 9: Thank you very much.

Point 10: Page 5, Line 210: Recommend the insertion of a hyphen between the words “small” and “scale” for improved accuracy. The new words should read “small-scale”

Response 10: Done. See Page 5, Line 228.

Point 11: Page 5, Line 214: I do not understand the term “teething problems” as it is used in this sentence. Recommend the authors replace this term with a more understandable term or phrase that better conveys what they seek to communicate.

Response 11: Thank you. The term has been changed to read’ ‘shortfalls related to’ on Page 5, Line 232.

Point 12: Page 6, Line 250: Recommend replacing the words “medicines use” with “the use of medicines” for improved understanding. The new sentence fragment should read “ . . . digital solutions to optimise the use of medicines by way of fostering . . . “

Response 12: Done. See Page 8, Line 310.

Round 2

Reviewer 1 Report

.